# Neuroprotective and Neurotrophic Potential of *Flammulina velutipes* Extracts in Primary Hippocampal Neuronal Culture

**DOI:** 10.3390/nu17193107

**Published:** 2025-09-30

**Authors:** Sarmistha Mitra, Raju Dash, Md Abul Bashar, Kishor Mazumder, Il Soo Moon

**Affiliations:** 1Department of New Biology, Daegu Gyeongbuk Institute of Science and Technology (DGIST), Daegu 42988, Republic of Korea; rajudash.bgctub@gmail.com; 2Department of Pharmacy, Faculty of Biological Sciences, Islamic University, Kushtia 7003, Bangladesh; bashar.iu.ph@gmail.com; 3Department of Pharmacy, Jashore University of Science and Technology, Jashore 7408, Bangladesh; kmazumder@just.edu.bd; 4Department of Anatomy, Dongguk University College of Medicine, Gyeongju 38066, Republic of Korea

**Keywords:** primary hippocampal neuron, neuritogenesis, neuroprotection, GC-MS, NTRK receptor pathway, flammulina velutipes metabolites

## Abstract

*Flammulina velutipes* (enoki mushroom) is a functional edible mushroom rich in antioxidants, polysaccharides, mycosterols, fiber, and minerals. Accumulating evidence highlights its therapeutic potential across diverse pathological contexts, including boosting cognitive function. However, its role in neuromodulation has not been systematically explored. This study examined the effects of methanolic and ethanolic extracts of *F. velutipes* on primary hippocampal neurons. Neurons were treated with different extract concentrations, followed by assessments of cell viability, cytoarchitecture, neuritogenesis, maturation, and neuroprotection under oxidative stress. The extracts were further characterized by GC-MS to identify bioactive metabolites, and molecular docking combined with MM-GBSA binding energy analysis was employed to predict potential modulators. Our results demonstrated that the methanolic extract significantly enhanced neurite outgrowth, improved neuronal cytoarchitecture, and promoted survival under oxidative stress, whereas the ethanolic extract produced moderate effects. Mechanistic studies indicated that these neuroprotective and neurodevelopmental benefits were mediated through activation of the NTRK receptors, as validated by both in vitro assays and molecular docking studies. Collectively, these findings suggest that *F. velutipes* extracts, particularly methanolic fractions, may serve as promising neuromodulatory agents for promoting neuronal development and protecting neurons from oxidative stress.

## 1. Introduction

Healthy functional foods with rich nutritional value are ideal for high-quality lifestyles and food habits. As people focus more on adopting healthier lifestyles and consuming nutritious organic foods, functional food is becoming increasingly popular, and awareness of it is being raised. Besides providing high nutrition, metabolites from functional foods can help in ailments of many diseases and health problems. Especially in the case of healthy aging, mushrooms work as an anti-aging super-food, which inhibits ROS accumulation with aging [1]. Mushrooms are a popular healthy choice of functional food because of their low carbohydrate content, high protein, and fiber. Also, many healthy food metabolites with medicinal properties, like ergosterol, polyphenol, and tannin, are enriched in mushrooms [2]. According to the Department of Agriculture, the United States, mushrooms are categorized as “other vegetables” and are generally considered plant-sourced food [3,4]. Among the many health benefits of mushrooms, improving brain function and boosting memory are interesting factors that draw the attention of research studies and open a new window for drug design and therapeutic applications.

*Flammulina velutipes*, commonly known as “golden needle mushroom” or “enoki mushroom,” is a common and widely consumed mushroom, especially in the Asian region [5]. Because of its high content of fiber, minerals, polysaccharides, and other healthy metabolites [6], this mushroom is a healthy choice. Apart from its traditional food value, it is also reported to have some medicinal properties. As it has a higher content of antioxidant agents, it protects neurons from damage and boosts brain function [7,8,9]. Recently, a neuroprotective peptide has been isolated from this mushroom, which can improve cognitive function [10]. Neurotropic natural compounds and brain-boosting agents have a very high therapeutic impact in the case of memory, as well as age-related neurodegenerative diseases. Memory loss, neurodegeneration, and cognitive dysfunction are some of the common phenomena in aging people [11,12,13] that degrade their quality of life and generate a significant burden on health care systems [14,15]. Different neurodegenerative diseases like Alzheimer’s disease, Parkinson’s disease, Amyotrophic lateral sclerosis, multiple sclerosis, and Huntington’s disease are common examples among the aging population, affecting cognitive function and memory [16,17]. Besides medicine, functional foods and alternative natural therapies can help manage cognitive dysfunction and memory loss [18]. Previous research studies have shown that *F. velutipes* mushroom is a natural memory-boosting and cognition-enhancing agent [8,9,10,19]. It has been found that polysaccharides of *F. velutipes* show an acetylcholinesterase inhibitory effect, which is responsible for its memory and cognition improvement effect [8].

In the present study, we evaluated the direct impact of ethanolic (FVEE) and methanolic (FVME) extracts of this mushroom on cultured neuron cells (Figure 1a). Firstly, we performed GC-MS analysis of both extract types to characterize the metabolites or chemicals present in the extract. We analyzed the effects of both types of extract on neurons, estimated and compared their effects on neuronal growth, development, and survival, and unleashed the mechanisms of their neuromodulatory effect. Our in vitro experimental assay and in silico molecular docking study suggested the active compound of the extract regulates neuronal growth via NTRK receptors. Our analysis revealed that the metabolites of the mushroom extracts function as neurotropic agents and promote neuronal growth by upregulating the neurotrophin signaling pathway.

## 2. Materials and Methods

### 2.1. Chemicals/Reagents

Media and all the supplements were bought from Invitrogen (Thermo Fisher Scientific, Waltham, MA, USA). Ethanol and methanol were bought from Sigma-Aldrich (Merck KGaA, St. Louis, MO, USA) for extract preparation. Linalool was purchased from MedChemExpress (Monmouth Junction, NJ, USA). and for use as a positive control in the experiments. Dil (1,1′-Dioctadecyl-3,3,3′,3′-Tetramethylindocarbocyanine Perchlorate) was purchased from Molecular Probes (Invitrogen, Eugene, OR, USA)

### 2.2. Preparation of Extract

Fresh fruiting bodies of *F. velutipes* were purchased from a local market in Gyeongju city/North Gyeongsang Province, South Korea. An author (Sarmistha Mitra) identified the species based on prior experience [20,21] with this species and examination of diagnostic macroscopic and microscopic characteristics, including cap morphology and color, gill attachment, stipe features, and spore size and shape. The species name was verified using MycoBank (https://www.mycobank.org, MycoBank No. 330940, accessed in 20 March 2021). A voucher specimen of *F. velutipes* has been deposited in the laboratory collection of the corresponding author and is available for future reference.

After cleaning and drying, the mushrooms were chopped and powdered using liquid nitrogen. The powdered mushroom material was soaked in separate containers in ethanol and methanol, and kept in a shaker for extract preparation. The powdered mushroom was filtered using filter paper. The dried concentrated extract was preserved at −20 °C, and a working extract of 8 mg/mL was prepared using dimethyl sulfoxide (DMSO) and further diluted with DMSO to reduce concentrations. The concentration–response curve was generated using the sum of all primary processes as a representative measure of morphological effects. Data for all tested concentrations (7.5–100 µg/mL) were shown, with the most effective concentrations (30, 60, 100 µg/mL) highlighted for subsequent experiments.

### 2.3. Standardization of the Extracts

Both the extracts were characterized by using GC-MS (gas chromatography–mass spectroscopy) to identify and quantify the active components in the extracts. The dried extracts were dissolved in DMSO to dissolve the most polar and nonpolar metabolites. The Agilent Technologies (Santa Clara, CA, USA) 7890A capillary gas chromatographic equipment was used for GC-MS analysis. The mass spectrometer detector on the chromatograph was composed of 95% dimethylpolysiloxane and 5% phenyl. A silica capillary column (film: 0.25 µm) with the brand and model number HP-5MSI served as the chromatographic column. It had a 0.25 mm diameter and a length of 90 m. The extract was injected into the GC apparatus in a fused silica capillary column in a total volume of 6 µL. The carrier gas (99.999% helium) was set to flow at a 1 mL/min rate. The temperature of the injector was kept at 250 °C, and the iron source temperature was adjusted to 280 °C. Initially, an isothermal temperature of 110 °C was maintained for 2 min. The temperature then rose by 10 °C/min until it reached 200 °C. The temperature then rose at a rate of 5 °C/min until it reached 280 °C, which was maintained for 9 min. The mass spectrum was collected in the mass-to-charge ratio region of 50 to 550 m/z. The ionization source was kept at a temperature of 250 °C, while the mass spectrometer quadrupole (MS quad) was kept at 150 °C. The full GC-MS analysis procedure required 36 min to complete, including the time for data gathering and the numerous temperature ramps and holds. During the study, mass spectra of unexplained peaks were gathered. The acquired mass spectra were compared to the NIST (National Institute of Standards and Technology, Gaithersburg, MD, USA) databases to determine the chemicals in the extract.

### 2.4. Primary Hippocampal Neuronal Culture

Primary hippocampal culture was performed in time-pregnant Sprague-Dawley rats (19th day), which were accommodated according to the Principles of Laboratory Animal Care (NIH, Washington, DC, USA). The whole culture procedures followed the Principles of Laboratory Animal Care and were approved by the Institutional Animal Care and Use Committee of Dongguk University (approval certificate number IACUC-2023-07) on 4 May 2023. The fetal hippocampi were collected from the brain, and dissociated neuronal cultures were prepared. The dissociated cells were seeded onto 12 mm coverslips previously coated with poly-DL-lysine (Sigma-Aldrich, Merck KGaA, St. Louis, MO, USA) in 24-well culture plates at a 1.0  ×  104 cells/cm^2^ density. Media for neuronal cells were prepared using Neurobasal media supplemented with B27, glutamate, and β-mercaptoethanol, kept in the incubator at 37 °C, with 5% CO_2_ and 95% air. Different extracts and/or vehicle concentrations were added to the culture plates before cell seeding. Also, linalool (50 nM) was used to compare the neuronal growth mediated by the extracts.

### 2.5. Hypoxia/Reoxygenation (H/R)-Induced Oxidative Stress

During culture, hippocampal cells were fed every 4 days by replacing 1/4 of the media with fresh prewarmed serum-free neurobasal media supplemented with B27, with or without sample extract. Hypoxia/reoxygenation (H/R) of rat hippocampal neurons was carried out, as described previously [22,23]. Briefly, after growing the cells for 10 days, the cells were transferred into a modular hypoxia chamber (Modular Incubator Chamber MIC-101; Billups-Rothenberg, Inc., Del Mar, CA, USA) in a 94% N2, 5% CO_2_, and 1% O_2_ atmosphere and incubated at 37 °C for 4 h. To reoxygenate hypoxic cells, culture plates were placed in a normal cell incubator (95% air, 5% CO_2_) and reoxygenated for 3 days. Then, neuronal viability was measured using the trypan blue assay. Prior to H/R treatment, cells received one of the following treatments: control, H/R stress without extracts (FVME/FVEE), or H/R stress with 30, 60, and 100 µg/mL concentrations of both FVME/FVEE for two hours.

### 2.6. Trypan Blue Exclusion Assay for Neuronal Viability

The trypan blue exclusion assay revealed the viability of neurons after treatment with different concentrations of extracts and after the H/R test. Cells were stained with 0.4% trypan blue solution for 30 min and washed with D-PBS solution before imaging. Phase-contrast images of all treatment groups were taken. The dead cells appeared dark blue; however, the viable cells appeared normal because they excluded the dye.

### 2.7. Immunocytochemistry

Immunocytochemistry was carried out in the same way as described in earlier works. After culturing the neurons on the coverslips for the indicated time, the neurons on the coverslips were washed briefly with D-PBS, fixed with a 4% paraformaldehyde and methanol fixation procedure, and blocked with 1% goat serum. Neurons were then incubated with primary antibodies overnight, followed by secondary antibody incubation for 2.5 h, and mounted on slides.

The following antibodies were used for immunostaining: primary antibodies of tubulin α-subunit (mouse monoclonal 12G10, 1:25; Developmental Studies Hybridoma Bank, University of Iowa, Iowa City, IA, USA), Ankyrin G (rabbit polyclonal H-215, 1:25; Santa Cruz Biotechnology Inc., Santa Cruz, CA, USA), MAP2 (mouse monoclonal clone HM-2; 1:250; Sigma, St. Louis, MO, USA), and Tau (rabbit polyclonal LF-PA0172; 1:100; AbFrontier, Seoul, Korea) and secondary antibodies (Alexa Fluor 488-conjugated goat anti-mouse IgG [1:500], Alexa Fluor 568-conjugated goat anti-rabbit IgG [1:1000], and Alexa Fluor 568-conjugated rabbit anti-goat IgG [1:1000], Molecular Probes, Eugene, OR, USA).

Neurones at the DIV3 were fixed and stained for 20 min at room temperature using Dil dye (20 μg/mL). The cells are rinsed three times with D-PBS after the incubation period.

### 2.8. Image Analysis and Quantification

Cells were fixed and processed for cytochemistry as described in Section 2.7. Imaging was performed using a Leica DM IRE2 research microscope (Leica Microsystems AG, Wetzlar, Germany) equipped with I3 S, N2.1S, and Y5 filter sets. Phase-contrast and epifluorescence images (1388 × 1039 pixels) were acquired with a high-resolution CoolSNAP™ CCD camera (Photometrics, Tucson, AZ, USA) under the control of Leica FW4000 software. Image acquisition parameters, including objective lens magnification, illumination settings, and exposure times, were constant across all experimental groups.

Morphometric analyses were conducted using ImageJ (v1.49) with the Simple Neurite Tracer plugin. Neurons were randomly selected, avoiding overlapping or truncated cells. Each cell was traced from the soma to the tips of all visible neurites, and the following features were quantified: axon length, the number of dendrites, dendritic length, axonal and dendritic branch number and length, and the number of branching points. Sholl analysis was performed using the Sholl analysis plugin (v1.0) with concentric circles spaced at 10 μm intervals from the soma center. Incomplete or overlapping neurites were excluded from the analysis.

For each experimental condition, 30 neurons from at least three independent cultures were analyzed. Digital measurements were exported and subjected to statistical analysis. Data normality was assessed using the Shapiro–Wilk test, and parametric and non-parametric tests were applied as appropriate. Results are presented as the mean ± SD. All acquisition, tracing, and analysis procedures were performed consistently to ensure reproducibility.

### 2.9. In Silico Identification of Potential Bioactive Compounds

#### 2.9.1. Glide-Based Molecular Docking Analysis

To extract the chemical structures of the compounds found in the GC-MS results, we used the PubChem database. While we found a total of 21 and 22 compounds in ethanol and methanol extracts, respectively, we considered compounds for molecular docking, excluding duplicates and solvent compounds. Additionally, we retrieved two compounds considered as controls, 3β, 6β-dichloro-5-hydroxy-5α-cholestane and 7,8-dihydroxyflavone for NTRK1 (also called TrkA) and NTRK2 (also referred to as TrkB), respectively [24,25]. The LigPrep module of the Schrödinger suite 2023-2 was harnessed to construct three-dimensional (3D) representations of these structures. By adjusting the pH to 7.0 and employing the Epik 2.2 program, the ionization states of each ligand and control were predicted [25]. This study created a maximum of 32 probable conformers for each ligand and control, and from these, the conformer with the least energy was selected for additional analysis.

The 3D crystal structures of NTRK1 and NTRK2 bearing PDB IDs 1WWW (NTRK1) and 1HCF (NTRK2) were extracted from the RCSB protein databank. These structures were subsequently prepared utilizing the protein preparation module within the Schrödinger suite 2023-2 program, which involved assigning the appropriate charges, hydrogen, and bond orders to 3D structures. All hydrogen bonds (H-bonds) within the crystal structure were optimized at a neutral pH by removing irrelevant water molecules.

The procedure of minimization was carried out by utilizing the OPLS_4 force field, taking into consideration that the structural modification should not have exceeded 0.30 Å of its root mean square deviation. By constructing a grid box at the reference ligand binding site of the protein, the active site of the protein was set for docking simulation. The parameters of grid generation remained at their default settings, with a bounding box size of 15 Å × 15 Å × 15 Å, and post-minimization was carried out harnessing the OPLS_4 force field. The van der Waals scaling factor and charge cutoff were fixed at 1.00 and 0.25, respectively [26]. With the aid of Schrodinger-Maestro version 9.4 [27,28], glide docking via extra precision (XP) was executed, where all ligands and controls were regarded as flexible by considering the van der Waals factor (0.80) and the partial charge (0.15), and the OPLS_4 force field was employed to minimize the docked complex. Following docking, for each ligand and control, the docking pose with the lowest glide score was considered for further analysis [26].

#### 2.9.2. Binding Energy (MM-GBSA) Analysis

Next, binding free energy (MM-GBSA) analyses were carried out harnessing the Prime MM-GBSA wizard of the Schrödinger suite 2023-2 program [29], in which an increased negative value indicates higher stability. The docking pose viewer file with the lowest glide score obtained following glide XP docking was subjected to analyses, in which the sampling minimization protocol was implemented with the OPLS2005 force field as Molecular Mechanics (MM) and Generalized Born Surface Accessible (GBSA) as the continuum model, maintaining the flexibility of the protein [30,31,32,33]. The dielectric solvent model, VSGB 2.0, was employed to rectify the empirical functions of π-stacking and H-bond interactions [34].

### 2.10. Statistical Analysis

The findings of this study were reported using the mean ± standard deviation (SD), which was obtained from at least three unique replicates of each test, except when specified differently. For neuritogenesis analysis, each data point in the graph represents the mean of ≤10 neurons per replicate, while for early neuronal differentiation, calculations were based on 400–500 neurons per condition. With the assistance of GraphPad Prism, version 8.0.0 (San Diego, CA, USA), the normality of the datasets was assessed using the Shapiro–Wilk test. As all datasets passed the normality test (*p* > 0.05), parametric tests were applied. The statistical comparisons were carried out by employing either Student’s *t*-test or one-way analysis of variance (ANOVA), which was subsequently accompanied by Dunnett’s multiple-comparisons test. Results were considered statistically significant when the *p*-value was less than 0.05 (*p*-value < 0.05). Levels of significance are indicated as follows: * *p* < 0.05; ** *p* < 0.01; *** *p* < 0.001; **** *p* < 0.0001.

## 3. Results

### 3.1. Chemical Characterization of FVME and FVEE

From the GC-MS analysis of FVEE and FVME, we found 21 and 22 compounds, respectively (Appendix A). Among all these compounds, linalool and dodecane, 1-fluoro- were common in both the extracts. GC-MS analysis of FVEE showed a peak area of 0.089% with a retention time of 9.17 min for linalool. On the other hand, linalool in FVME was represented by a peak area of 0.2335% with the same retention time.

### 3.2. Effects of FVME and FVEE on Cell Viability

By performing the trypan blue exclusion assay, the effect of different concentrations of FVME and FVEE on neuronal viability was tested. Here, the numbers of live cells and dead cells were counted. The bright-field images are presented in Figure 1b. The statistical analysis of the data showed that (Figure 1b) all concentrations of the extracts enhanced the viability of the neurons.

### 3.3. Dose-Dependent Effects of FVME and FVEE on Neuronal Development

#### 3.3.1. FVME and FVEE Promote Neurite Outgrowth

From the morphological analysis performed on cultured hippocampal neuronal cells, we noticed a concentration-dependent growth effect of neurons due to treatment with FVME and FVEE. Cultured neurons were allowed to grow until the third day, with and/or without different concentrations of FVME and FVEE (Figure 2 and Appendix A). On the third day, phase-contrast images were taken for performing analysis. We analyzed morphological parameters of neuronal growth, including the number of primary processes, the length of the longest process, and the total length of all processes, as shown in Figure 2b,c. According to all the parameters, the neurons showed significant growth in the extract-treated condition compared to the control (vehicle) condition. Fixed neurons (DIV3) were stained with Dil dye and are represented in Figure 2a. Based on the results, we used 30 µm FVME and 100 µm FVEE concentrations for further analysis (Appendix A).

However, the neuronal growth on FVME-treated neurons was more significant than that on FVEE-treated neurons. FVME treatment significantly improved all three growth parameters (Figure 2b). This implies that FVME has a more prominent effect on neuronal growth than FVEE. Based on this finding, we performed further neuro-architecture analysis by treating neurons with FVME.

#### 3.3.2. FVME and FVEE Promote Early Neuronal Development

After investigating the optimized concentrations of FVME and FVEE, we applied these concentrations to primary cultured neurons to investigate their impact on early development. Neuronal maturation or early development of neurons was analyzed by incubating the neurons with vehicles and the extracts at the mentioned concentrations until the first and second day of culture (Figure 3a,d). The developmental phases of neurons were divided into three categories: lamellipodia, minor processes, and axonal outgrowth (Figure 3b,e) [35]. Our statistical analysis (Figure 3c,f) showed that after treatment with FVME and FVEE, more neurons were transformed into the advanced developmental stage compared to those with the vehicle treatment. During the first 24 h, both the extracts showed significant advancement in developing the neurons into the minor process stage compared to the vehicles. At 48 h, the neuron’ development into the axonal outgrowth stage was significantly promoted by both extracts. Neurons at 24 h and 48 h, with or without extract treatment, were immunostained with Tau and Map2 antibodies (Figure 3a,d), which are axonal and dendritic markers, respectively. Enlarged in-sets are represented in Figure 3b,e. In this experiment, 30 µM of FVME and 100 µM of FVEE were used.

#### 3.3.3. Axonal Development Due to FVME

Because the FVME showed a prominent neuronal development effect, we explored the effect of FVME (100 μg/mL concentration) on neuronal branching by “Sholl analysis”. The morphological analysis showed that FVME significantly improved the axonal growth outcome in cells growing until the fifth day of the culture. The analysis showed FVME significantly improved axon length (Figure 4b) and branching of neurons (Figure 4c). As represented by the statistical analysis (Figure 4c), FVME significantly increased the number and length of primary, secondary, and tertiary branches of axons. Neurons immunostained with the axonal marker Ankyrin G and microtubule marker α-Tubulin are represented in Figure 4a.

For more detailed insight, we performed “Sholl analysis”, which showed that FVME treatment increased axonal intersection and collateral branching, which eventually improved the cytoarchitecture of the neurons (Figure 4d,e).

Neurons treated with FVME had axonal intersections increase by 1.64-fold compared to vehicle-treated neurons (Figure 4d). Again, neurons treated with FVME showed axonal intersection up to the circle of 220 μm, while control neurons exhibited it only up to the 190 μm concentric Sholl circle. Furthermore, FVME increased the collateral branching points by 3.04-fold compared to the vehicle (Figure 4e), and thus, collateral branching points for FVME-treated neurons could be observed up to the 270 μm Sholl circle, while there were none beyond 120 μm in the vehicle treatment.

#### 3.3.4. Dendritogenic Arborization Due to FVME

Dendritic branching is a crucial factor for memory storage. So next, we analyzed the dendritic development after FVME treatment and incubation until DIV5 (Figure 5). The morphological analysis showed that FVME treatment promoted neurons to upregulate the number of dendrites and branches (Figure 5a,b). Firstly, the number and length of primary dendrites were significantly increased by FVME treatment (Figure 5c). Secondly, the number and length of branches (primary and secondary) of dendrites were elevated significantly (Figure 5d).

Later, Sholl analysis also revealed an improvement in dendritic arborization. Treatment with FVME enhanced dendritic intersection 1.63-fold compared to the vehicle (Figure 5e). Also, dendritic branching points were increased 3.5-fold compared to those under the vehicle (Figure 5f). Altogether, this data suggests that FVME promotes dendritic development.

### 3.4. FVME and FVEE Protect Neurons Against H/R Injury

To evaluate the neuroprotective effect of FVME and FVEE, we induced hypoxia-mediated oxidative stress and cellular injury. Neurons are especially susceptible to hypoxia, as the absence of oxygen causes neuronal injury, initiates neuronal death, stops neuronal function, and restricts synaptic transmission [36]. Our experiment induced hypoxia for 4 h and analyzed the viability by trypan blue assay after 4 days of hypoxia.

Figure 6a,b shows that FVEE significantly increased neuronal viability compared to the vehicle treatment. FVEE at a 100 µM concentration significantly attenuated hypoxia-mediated neuronal death. In the case of FVME, all concentrations showed a significant neuroprotective effect compared to the vehicle (Figure 6c,d). However, the 60 µM concentration showed the best neuroprotective effect based on neuronal viability. These outcomes suggest that both FVME and FVEE can provide neuroprotection against hypoxia-induced brain injury and neuronal death.

### 3.5. Mechanistic Analysis of the Neurogenic Effect of FVME and FVEE

Neuritogenesis is facilitated by neurotropic signaling pathways, wherein neurotrophins activate NTRK receptors and initiate downstream cascades that govern axonal and dendritic growth [37]. Furthermore, by activating NTRK receptors, this pathway is also implicated in neuroprotection by initiating neuronal survival pathways and inhibiting apoptosis [38,39,40]. Therefore, we explored whether the neurotropic signaling pathway is responsible for the neurotrogenic activity of FVME and FVEE. Using both NTRK1 and NTRK2 receptor inhibitors, we analyzed neuronal morphological growth and found neuronal growth inhibition in the presence of the inhibitors. We used GW441756 and Ana 12 as NTRK1 and NTRK2 inhibitors, respectively. Figure 7 shows neuronal growth restriction by inhibition of NTRK receptor subtypes. The representative images show neurons treated with the extract in the presence or absence of inhibitors (Figure 7a). The reduction in the morphological parameters (the sum of all the processes, number of primary processes, and length of the longest process) illustrates evidence that FVME and FVEE exert neurotropic action via the NTRK signaling pathway (Figure 7b–d). Statistical analysis showed that FVEE-mediated neuronal growth depends on NTRK1 and NTRK2 receptors, as both inhibitors downregulated the growth. Nevertheless, in the case of FVME, the growth parameters were mainly inhibited by Ana-12 (NTRK1 inhibitor), indicating NTRK1 receptor-mediated neurotropic action. We used 100 µM of FVEE and 60 µM of FVME.

### 3.6. In Silico-Based Identification of Potential NTRK Receptor Modulators

Next, we further assessed the binding efficiency of the compounds identified from methanol and ethanol extracts of *F. velutipes* by GC-MS, for two potential targets, NTRK1 and NTRK2, using molecular docking analysis and binding energy calculation. At first, we conducted molecular docking using an extra-precision algorithm, followed by MM-GBSA binding energy calculation, which revealed that all compounds exhibited binding energy ranging from −9 kcal/mol to −38 kcal/mol with the NTRK1 receptor and from −10 to −43 kcal/mol with the NTRK2 receptor (Figure 8a and Appendix A). Tricyclo [3.3.0.0(2,8)]octan-3-one, 4-[2-(m-anisyl)ethyl]-8-methyl- and benzenemethanol, .alpha.-(2-aminocyclopentyl)- identified in methanol extracts had the highest binding energy scores with NTRK1 and NTRK2 receptors, where the binding energies of tricyclo [3.3.0.0(2,8)]octan-3-one, 4-[2-(m-anisyl)ethyl]-8-methyl- with NTRK1 and NTRK2 were −36.9 kcal/mol and −30.3 kcal/mol, respectively, whereas the binding energies of benzenemethanol, .alpha.-(2-aminocyclopentyl)- were −36.2 kcal/mol and −32 kcal/mol, respectively (Appendix A). In contrast, in ethanolic extracts, hentriacontane and 5-acetoxymethyl-2,6,10-trimethyl-2,9-undecadien-6-ol had the highest binding energy scores, and Appendix A demonstrates that the binding energies of hentriacontane with NTRK1 and NTRK2 were −38.2 kcal/mol and −43.7 kcal/mol, whereas the binding energies of 5-acetoxymethyl-2,6,10-trimethyl-2,9-undecadien-6-ol were −37.1 kcal/mol and −37.7 kcal/mol, respectively.

The controls, 3β, 6β-dichloro-5-hydroxy-5α-cholestane and 7,8-dihydroxyflavone, had binding energies of 34 kcal/mol and 30.9 kcal/mol against NTRK1 and NTRK2 receptors, respectively (Figure 8a). As shown in Figure 8b, the control, 3β, 6β-dichloro-5-hydroxy-5α-cholestane formed one H-bond with NTRK1 with the residue Gln350 and several hydrophobic interactions with Phe317, Leu322, Phe327, Thr352, Val354, and Asn355, whereas the other control, 7,8-dihydroxyflavone, formed double H-bonds with NTRK2 with the residue Asn350 and some hydrophobic interactions with Tyr319, Tyr329, and Ile330. Two compounds, linalool and dodecane, 1-fluoro-, were found common in both methanol and ethanol extracts, where linalool had binding energies of −23 kcal/mol and −23.2 kcal/mol with NTRK1 and NTRK2 receptors, respectively, whereas dodecane, 1-fluoro- showed binding energies of −20.3 kcal/mol and −25.5 kcal/mol, respectively. However, their binding energies with NTRK1 and NTRK2 were notably lower (Figure 8a). Since both compounds were found in both methanol and ethanol extracts, we analyzed their interaction profiles, which showed that both compounds formed interactions with several residues, including a few residues that formed bonds with controls (Figure 8b). As shown in Figure 8b, linalool formed a single hydrophobic interaction with Phe317 with NTRK1 receptors, whereas it formed multiple H-bonds with Val321 and Asn323. On the other hand, with NTRK2, linalool formed dual hydrophobic interactions with Tyr319 and Leu324 and double H-bonds with Ile323 and Ans325 (Figure 8b). Dodecane, 1-fluoro- formed several hydrophobic interactions with NTRK1 receptors with Phe327, Thr352, and Val354 residues. In contrast, dodecane, 1-fluoro- formed many hydrophobic interactions with NTRK2 receptors with Tyr319, Leu324, Tyr329, and Ile330 residues (Figure 8b).

We further tested the neuritogenic effect of linalool on primary culture hippocampal neurons in comparison to our experimental extracts, as linalool is a well-established compound that is involved in treating neurodegeneration [41,42,43] and has neuroprotective [42,44] and memory-boosting activity [45]. As shown in Appendix A, linalool significantly increased both the number and the total length of primary neurites compared to the control. In contrast, FVME significantly enhanced all measured parameters relative to the control and linalool.

## 4. Discussion

Neuroprotective and brain-boosting neurotrophic compounds found in natural sources can be used as future medicinal candidates for managing neurodegeneration. Loss of neurotrophic support is a hallmark of neurite degeneration leading to neurodegenerative disorders (NDs). Thus, pharmacological agents that promote neuritogenesis may offer therapeutic potential by reversing early pathological changes.

In this study, we demonstrated the efficacy of FVEE and FVME in promoting early development and differentiation of primary hippocampal neurons. From the initial viability, maturation, and neuro-morphological analysis, we found that FVME and FVEE initiate neuronal viability, neurite outgrowth, and network formation, indicating the presence of neuro-developmental bioactive compounds with both neurotrophic and neuroprotective properties. However, neuromorphological analysis, early developmental analysis, and viability studies show that FVME has more neurodevelopmental potential than FVEE.

Across early stages (Stages I–II), both FVME and FVEE significantly promoted neurons from the early developmental phase (lamellipodia) to the mature phases (neurons with minor processes and axonal outgrowth). In both neuronal growth analysis and early development, FVME showed more prominent action than FVEE; therefore, we further analyzed the effect of FVME on mature neurodevelopmental stages. In the mature stage, axonal and dendritic extension and branching are key parameters for establishing robust neural networks. These morphological enhancements include improved synaptic plasticity and cognitive function [46,47]. According to our experimental findings, FVME significantly upregulated axonal and dendritic arborization and architecture (Figure 4 and Figure 5). Upregulated axonal and dendritic arborization enhances neuronal connectivity by promoting synaptic architecture and network integration, thus implicating cognitive enhancement and memory function [48,49,50].

We further performed GC-MS analysis of ethanolic and methanolic extracts to characterize the key metabolites and the potential bioactive compounds promoting *F. velutipes*-mediated neuronal growth. GC-MS analysis identified 21 and 22 metabolites in FVEE and FVME, respectively, which were subsequently subjected to molecular docking against NTRK receptors. The docking results indicated multiple bioactive compounds from both extracts exhibiting high binding affinity toward NTRK receptors. Among the metabolites common to both extracts, linalool was selected for further validation in primary hippocampal culture. Linalool treatment elicited a significant neurotropic response, suggesting that the neurotropic activity of FVME and FVEE is partially attributable to linalool (Appendix A). Furthermore, compared to FVME, the effect of linalool was less effective, which may be explained by synergistic interactions among multiple bioactive constituents of FVME, conferring a broader and stronger neuritogenesis effect than linalool alone.

GC-MS analysis identified linalool in both FVME and FVEE at a retention time of 9.17 min, consistent with previous findings [51,52], and this was reported to be detected by a similar thermal decomposition used in other studies [53,54,55]. The relative abundance was higher with FVME (0.2335% peak area) than with FVEE (0.089%). Linalool is a monoterpene volatile compound mostly available in aromatic plants [56], and many studies have established linalool as a potential phytochemical for treating neuronal diseases [42,57,58]. The binding analysis study with NTRK receptors suggested that linalool could bind effectively with NTRK receptors. Our molecular docking and binding energy calculation revealed that linalool effectively binds to both NTRK receptors. These interactions included residue Phe317 in NTRK1 and Tyr319 in NTRK2, which were also observed in controls (Figure 8b). The observed interaction between control and Phe317 in NTRK1 aligns with previous findings. Additionally, the control also formed further interactions within the NTRK1 at residues Leu322, Phe327, Gln350, Thr352, Val354, and Asn355 (Figure 8b), all of which have been documented in prior research [25].

Our findings highlight the NTRK signaling pathway-dependent neurodevelopmental mechanism through which FVME and FVEE promote neuronal growth, axonal and dendritic sprouting, and survival (Figure 9). While neurotrophin signaling appears to be a primary mediator, we do not exclude the possibility that the extracts modulate additional pathways through their diverse chemical constituents. A limitation of our study is that the experiments were conducted exclusively in vitro using crude extracts and linalool, which may not fully reflect the complexity of in vivo neuronal environments. Moreover, while GC-MS allowed us to identify several metabolites, thermal decomposition cannot be excluded, and other major bioactive compounds besides linalool remain to be investigated. Finally, the lack of standardized positive controls limits the extent to which the neurotrophic potency of FVME and FVEE can be directly compared with known pharmacological agents. These results position *F. velutipes* extracts, particularly the methanolic fraction, as a promising natural source of neurotrophic compounds, as suggested by in silico molecular docking and binding energy calculations. Future in vitro and in vivo studies and detailed characterization of additional metabolites will be critical to translate these findings into potential therapeutic strategies for neurodegenerative disorders.

## 5. Conclusions

In this present work, we experimentally proved the role of methanolic and ethanolic extracts of *F. velutipes* mushroom as neurotrophic agents. Our in vitro analysis revealed that both extracts of *F. velutipes* mushroom have significant neuronal developmental and neuroprotective effects that can be useful in treating neurodegenerative and memory-related disorders. Altogether, this mushroom has the potential to be used as a medicinal agent for modulating the NTRK-dependent signaling pathway. Additional neuro-morphological and in silico molecular docking analysis also proved that bioactive metabolites of these extracts interact with NTRK receptors. These findings suggest that *F. velutipes* should be studied more and analyzed mechanistically for future drug design for neurodegenerative diseases and dementia.

## Figures and Tables

**Figure 1 nutrients-17-03107-f001:**
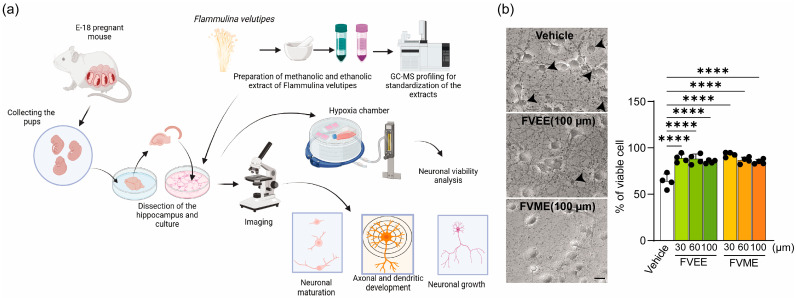
FVME and FVEE preserve the viability of neurons. (**a**) represents the overall experimental plan from primary neuronal culture to data analysis. (**b**) shows bright-field images of neurons grown until DIV8 and stained with trypan blue after being treated in the vehicle/FVEE/FVME condition. The dead neurons are marked with black arrows. Statistical analysis represents the viability of neurons with different doses of FVEE and FVME. Data are presented as mean ± SD from three independent biological replicates, with individual data points shown. Statistical significance was assessed by one-way ANOVA followed by Dunnett’s multiple comparisons test versus vehicle (**** *p* < 0.0001). Scale bar: 20 µm. FVME, *Flammulina velutipes* methanolic extract; FVEE, *Flammulina velutipes* ethanolic extract.

**Figure 2 nutrients-17-03107-f002:**
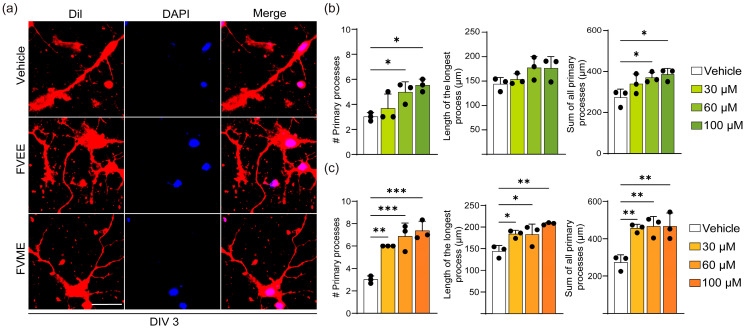
Neurons’ morphological growth parameters significantly improved due to FVME and FVEE. Neuronal growth was observed on the third day of primary culture and stained with Dil dye (red), followed by counterstaining with DAPI (blue). (**a**) Representative fluorescent image of neurons grown with normal media/FVEE/FVME. Statistical analysis represents morphological parameters, the number of primary processes, the length of the longest process, and the sum of all the processes in neurons with different doses of FVEE (**b**) and FVME (**c**). Data are presented as mean ± SD from three independent biological replicates, with individual data points shown. Statistical significance was assessed by one-way ANOVA followed by Dunnett’s multiple comparisons test versus vehicle. * *p* < 0.05; ** *p* < 0.01; *** *p* < 0.001; scale bar = 20 µm. FVEE, *Flammulina velutipes* ethanolic extract; FVME, *Flammulina velutipes* methanolic extract; Dil, 1,1′-Dioctadecyl-3,3,3′,3′-Tetramethylindocarbocyanine Perchlorate.

**Figure 3 nutrients-17-03107-f003:**
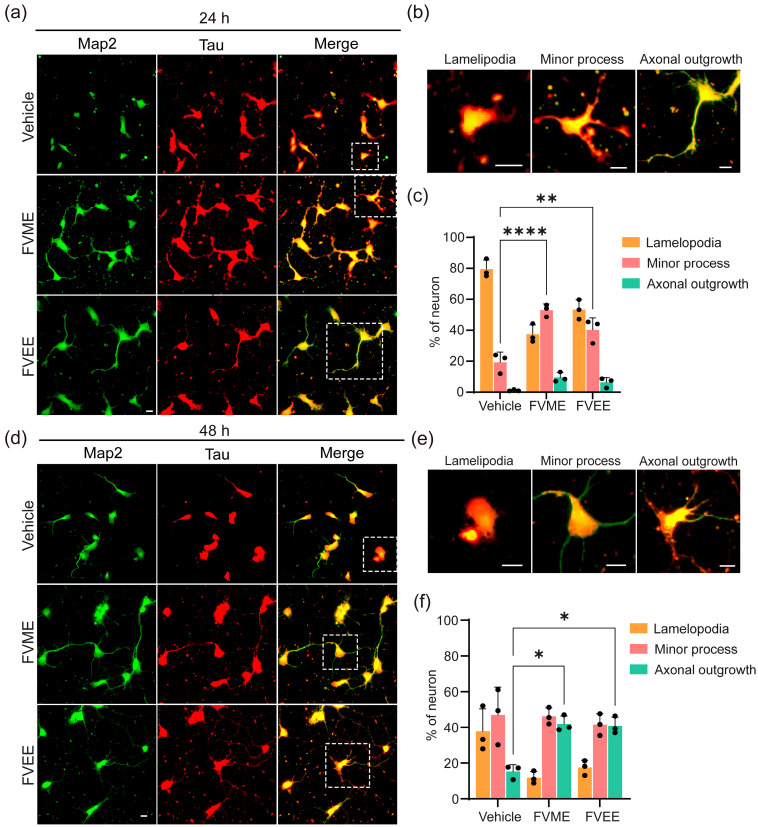
FVME and FVEE significantly promote neuronal maturation in the early stage of neuronal development. (**a**) Neuronal growth in the vehicle/FVME/FVEE-treated conditions at 24 h, immunostained with Map2 (neuronal marker, green) and Tau (dendritic marker, red). Representative images showing neuronal growth stages (lamellipodia, minor process, axonal outgrowth) at 24 h (**b**) and the corresponding statistical analysis (**c**). (**d**) Microscopic images showing the neuronal growth in the vehicle/FVME/FVEE-treated conditions at 48 h, immunostained with Map2 (green) and Tau (red). Representative images show neuronal growth stages (lamellipodia, minor process, axonal outgrowth) at 48 h (**e**) and the corresponding statistical analysis (**f**). Each statistical analysis presents data as mean ± SD from three independent biological replicates, with individual data points shown. Statistical significance was assessed by one-way ANOVA followed by Dunnett’s multiple comparisons test versus vehicle (* *p* < 0.05; ** *p* < 0.01; **** *p* < 0.0001). Scale bar: 10 µm. FVME, *Flammulina velutipes* methanolic extract; FVEE, *Flammulina velutipes* ethanolic extract.

**Figure 4 nutrients-17-03107-f004:**
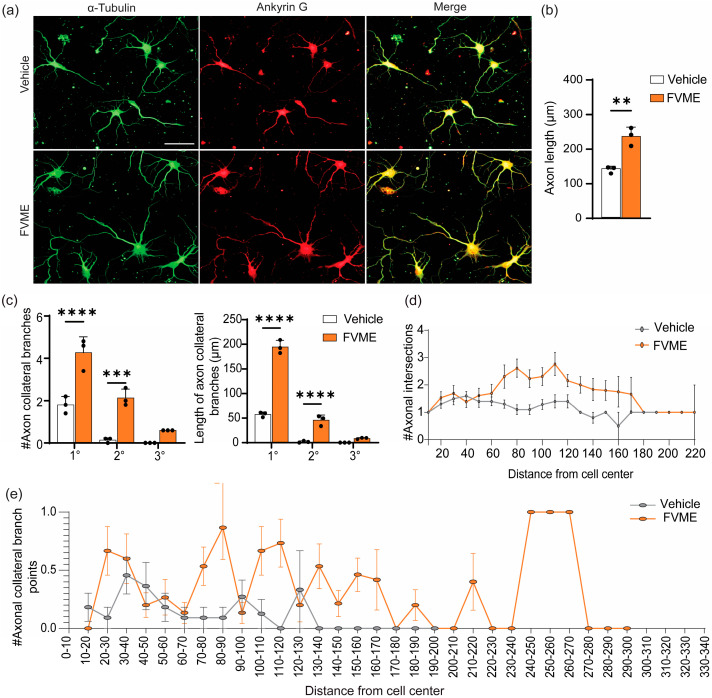
FVME promotes axonal growth significantly. (**a**) The primary cultured neurons were grown until DIV5. Neurons treated with the vehicle or FVME were immunostained with Ankyrin G (axonal marker, red) and α-Tubulin (microtubule marker, green). Morphological analysis showed that FVME significantly improved axonal length (**b**), axonal collateral branching (primary, secondary, and tertiary), and the length of axonal collateral branches (**c**). Sholl analysis of neurons revealed the number of axonal intersections (**d**) and axonal branching points (**e**). Data are presented as mean ± SD from three independent biological replicates, with individual data points shown. Statistical significance was determined using Student’s *t*-test (** *p* < 0.01; *** *p* < 0.001; **** *p* < 0.0001). Scale bar: 50 µm. FVME, *Flammulina velutipes* methanolic extract.

**Figure 5 nutrients-17-03107-f005:**
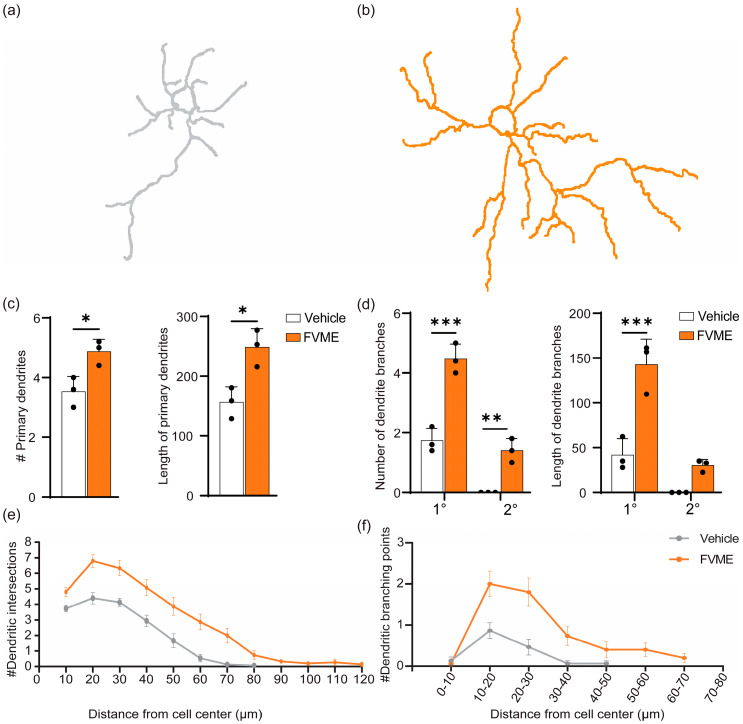
FVME upregulated dendritic arborization. Representative neurons traced for Sholl analysis are shown in (**a**,**b**), for the vehicle and FVME treatment, respectively. (**c**) Statistical analysis shows that FVME treatment significantly increased the number and length of primary dendrites. (**d**) Statistical analysis shows that FVME significantly increased the number and length of dendritic branches. Sholl analysis revealed numbers of dendritic intersections (**e**) and numbers of dendritic branching points (**f**), which were higher in the case of FVME treatment. Data are presented as mean ± SD from three independent biological replicates, with individual data points shown. Statistical significance was determined using Student’s *t*-test (* *p* < 0.05; ** *p* < 0.01; *** *p* < 0.001). FVME, *Flammulina velutipes* methanolic extract.

**Figure 6 nutrients-17-03107-f006:**
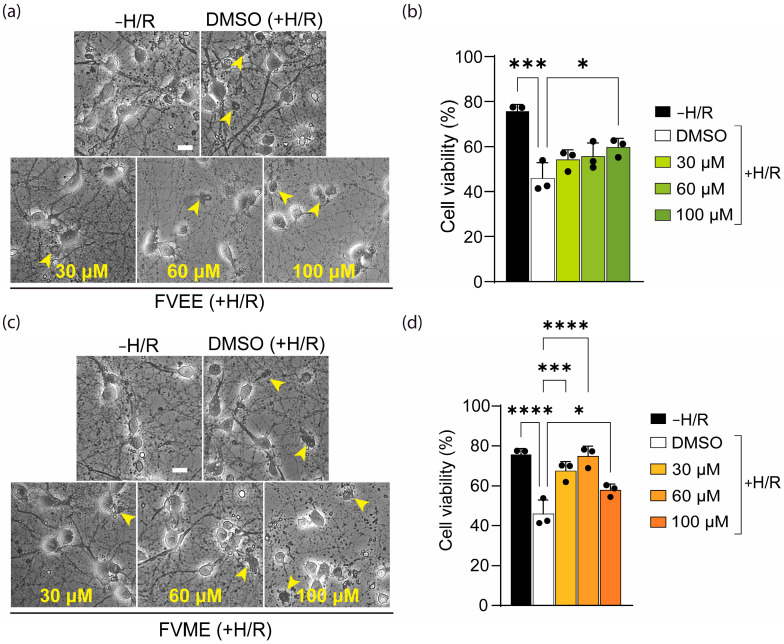
Neuroprotective effect of extracts (FVEE/FVME) on hypoxia-mediated neuronal death. (**a**) represents bright-field images of trypan blue-stained primary neurons grown until DIV13 in hypoxia/reoxygenation (H/R) conditions in the presence of the vehicle or various concentrations of FVEE. (**b**) The statistical analysis shows that various doses of FVEE improved neuronal viability; however, 100 µM significantly improved neuronal viability under H/R injury. (**c**) represents bright-field images of trypan blue-stained primary neurons grown until DIV13 in H/R conditions in the presence of the vehicle or various concentrations of FVME. (**d**) The statistical analysis shows that various doses of FVME (30, 60, 100 µM) improved neuronal viability significantly under H/R injury. The small arrow shows the dead cells in the culture. Data are presented as mean ± SD from three independent biological replicates, with individual data points shown. Statistical significance was by one-way ANOVA followed by Dunnett’s multiple comparisons test versus DMSO (* *p* < 0.05; *** *p* < 0.001; **** *p* < 0.0001). Scale bar = 20 µm. FVME, *Flammulina velutipes* methanolic extract; FVEE, *Flammulina velutipes* ethanolic extract.

**Figure 7 nutrients-17-03107-f007:**
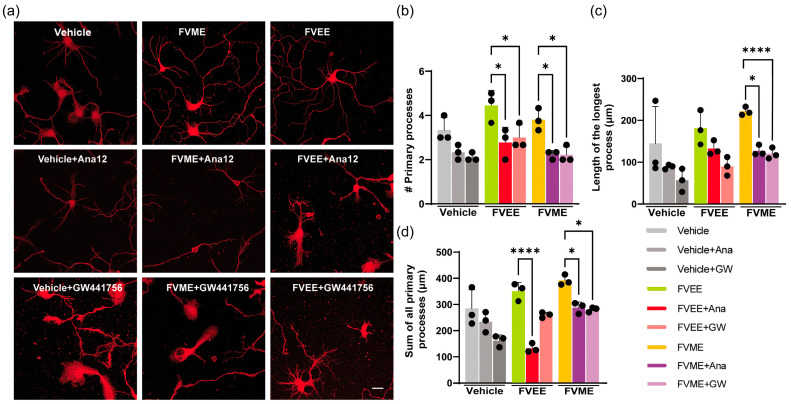
NTRK1 and NTRK2 receptors mediated neurodevelopment by FVME and FVEE. (**a**) represents neurons (DIV3) stained with tubulin in the absence or presence of the vehicle/FVME/FVEE with the NTRK1 inhibitor GW441756 and the NTRK2 inhibitor ANA12. Statistical analysis shows that neurodevelopmental parameters (the number of primary processes (**b**), length of the longest process (**c**), sum of all the processes (**d**) mediated by FVEE were inhibited by Ana12 and GW441756. FVME-mediated neuronal growth parameters were predominantly inhibited by Ana12. Scale bar = 20 µm. Data are presented as mean ± SD from three independent biological replicates, with individual data points shown. Statistical significance was determined by one-way ANOVA followed by Dunnett’s multiple comparisons test versus vehicle (* *p* < 0.05; **** *p* < 0.0001). FVME, *Flammulina velutipes* methanolic extract; FVEE, *Flammulina velutipes* ethanolic extract; NTRK1, Neurotrophic tyrosine kinase receptor type 1; NTRK2, Neurotrophic tyrosine kinase receptor type 2.

**Figure 8 nutrients-17-03107-f008:**
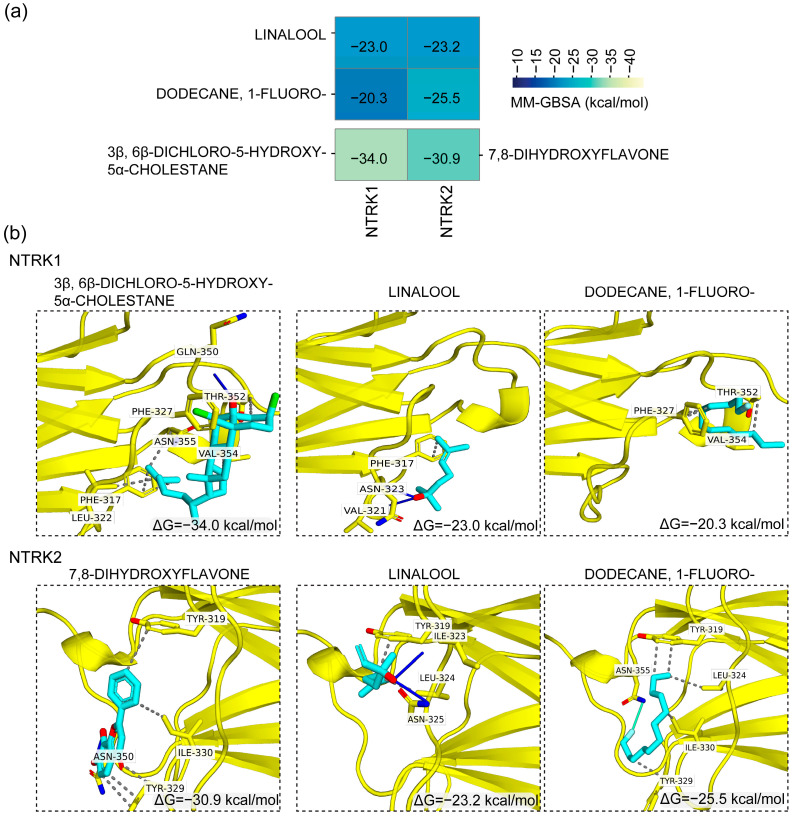
MM-GBSA binding energy-based screening and molecular interactions of compounds identified in methanol and ethanol extracts of *Flammulina velutipes*. (**a**) Heatmap of MM-GBSA binding energy of controls and compounds commonly found in methanol and ethanol extracts of *F. velutipes*. (**b**) Three-dimensional protein–ligand interaction profiling of controls and compounds from the methanol and ethanol extracts of *F. velutipes* with NTRK1 (top panel) and NTRK2 (bottom panel) receptors. The left panel displays 3D protein–ligand interactions of two controls, 3β, 6β-dichloro-5-hydroxy-5α-cholestane and 7,8-dihydroxyflavone, in the top and bottom panels, respectively, whereas the middle and right panels depict 3D protein–ligand interactions of compounds found in ethanolic and methanolic extracts of *F. velutipes*. In this case, the protein backbone and ligands are visualized using yellow and light blue colors, respectively, whereas the hydrophobic interactions and H-bonds are shown as dashed gray lines and solid blue lines, respectively. NTRK1, Neurotrophic tyrosine kinase receptor type 1; NTRK2, Neurotrophic tyrosine kinase receptor type 2.

**Figure 9 nutrients-17-03107-f009:**
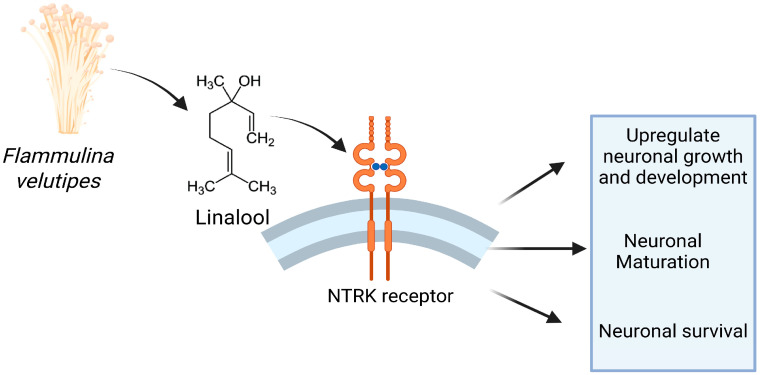
The pictorial representation shows that *Flammulina velutipes* modulates neuronal growth and viability in the NTRK-derived pathway. The NTRK receptors are activated by neurotrophic metabolites in the mushroom extracts and eventually promote neuronal maturation, morphological development, axonal and dendritic cytoarchitecture, and neuronal survival.

## Data Availability

Data presented in the manuscript will be available upon reasonable request.

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
