# Peer review of "Neuroprotective and Neurotrophic Potential of *Flammulina velutipes* Extracts in Primary Hippocampal Neuronal Culture"

_nutrients, 2025, doi:10.3390/nu17193107_

Round 1

Reviewer 1 Report

Comments and Suggestions for Authors

The title fails to encapsulate the core findings of the manuscript accurately. The abstract is shallow and needs revision to include results consistently; upon reading this section, I find the potential applications of the findings unclear. Additionally, the study's objective is not clearly articulated in the abstract. Provide concrete findings instead of generally citing the results. No proper methods were described. Keywords should be revised to utilize terminology distinct from both the title and the abstract. Please, improve it.  

Proper botanical identification must be provided to source the mushroom – “local market” is not an adequate source. The authors stated that “The mushrooms were characterized by expert personnel.”. Please, provide additional information about it. Still in section 2.2, there is an inadequate sentence: “Working extract of 8mg/ml was prepared. 30,60,100 μg/ml doses of both extracts were used in this study.” A concentration-response curve should be performed to comprehend the biological properties of the extract better. At least a rationale behind this concentration must be provided.

Revise all the text to replace the misuse of the word “dose” with “concentration,” considering the in vitro design of your study.

Section 2.5 – The treatments are not clear (concentration, time of incubation..).

Regarding the statistical test, no assessment of data normality was performed.

Figure 1C depicts the data from H/R-induced oxidative stress. The first bar refers to vehicle groups, but the viability is too low. No naïve group was used. Furthermore, all tested concentrations presented a similar effect, reinforcing the importance of a concentration-response curve.

Upon reviewing the results section, it became evident that the vast majority of the evaluations discussed lack proper documentation in the methodology (Sections 3.2 to 3.6). It is imperative to revise these sections to include a detailed account of the methods employed and to clarify the analytical approaches utilized.

Chemical characterization should be the first data presented for logical order.

In the discussion, authors stated that an in vitro test was also performed with linalool. But it was not properly described in the method section.

Figures are informative and of high quality.

The discussion lacks depth and essentially repeats the findings without adequately linking them to relevant literature. A more comprehensive analysis is needed. Additionally, it is recommended that the authors include a brief discussion on the potential limitations of their findings to allow for a more nuanced interpretation of the data.

Comments on the Quality of English Language

Revise all the documents to polish typographical errors. Misuse of abbreviations should be revised.

Author Response

Comment 1: The title fails to encapsulate the core findings of the manuscript accurately. The abstract is shallow and needs revision to include results consistently; upon reading this section, I find the potential applications of the findings unclear. Additionally, the study's objective is not clearly articulated in the abstract. Provide concrete findings instead of generally citing the results. No proper methods were described. Keywords should be revised to utilize terminology distinct from both the title and the abstract. Please, improve it.  

response:  Thank you for this insightful suggestion. We agree that the original title was too broad. To more accurately reflect our findings, we have revised the title to:
“Neuroprotective and Neurotrophic Potential of Flammulina velutipes Extracts in Primary Hippocampal Neuronal Culture.”

We have thoroughly revised the abstract to state the study objective clearly, describe the methods, present concrete results, and highlight the potential applications of our findings.

Flammulina velutipes (enoki mushroom) is a functional edible mushroom rich in antioxi-dants, polysaccharides, mycosterols, fiber, and minerals. Accumulating evidence high-lights its therapeutic potential across diverse pathological contexts, including boosting cognitive function. However, its role in neuromodulation has not been systematically ex-plored. This study examined the effects of methanolic and ethanolic extracts of F. velutipes on primary hippocampal neurons. Neurons were treated with different extract concentra-tions, followed by assessments of cell viability, cytoarchitecture, neuritogenesis, matura-tion, and neuroprotection under oxidative stress. The extracts were further characterized by GC-MS to identify bioactive metabolites, and molecular docking combined with MM-GBSA binding energy analysis was employed to predict potential modulators. Our results demonstrated that the methanolic extract significantly enhanced neurite out-growth, improved neuronal cytoarchitecture, and promoted survival under oxidative stress, whereas the ethanolic extract produced moderate effects. Mechanistic studies indi-cated that these neuroprotective and neurodevelopmental benefits are mediated through activation of the NTRK receptors, as validated by both in vitro assays and molecular docking studies. Collectively, these findings suggest that F. velutipes extracts, particularly the methanolic fraction, may serve as promising neuromodulatory agents for promoting neuronal development and protecting neurons from oxidative stress.”

Additionally, we have updated the keywords to better reflect the main findings while avoiding redundancy with the title and abstract.

Comment 2: Proper botanical identification must be provided to source the mushroom – “local market” is not an adequate source. The authors stated that “The mushrooms were characterized by expert personnel.”. Please, provide additional information about it.

Response:Thanks to the reviewer for raising the question. We added the modified version to section 2.2, line 85-92.

Comment 3: Still in section 2.2, there is an inadequate sentence: “Working extract of 8mg/ml was prepared. 30,60,100 μg/ml doses of both extracts were used in this study.” A concentration-response curve should be performed to comprehend the biological properties of the extract better. At least a rationale behind this concentration must be provided.

Response: According to the reviewer’s suggestion the concentration-response curve is represented in supplementary figure 1, showing why these concentrations were chosen. We initially applied concentrations ranged from 7. 5 to 100 ug/mL and chose the best three concentrations based on biological effect.

Comment 4: Revise all the text to replace the misuse of the word “dose” with “concentration,” considering the in vitro design of your study.

Response: Thanks to the reviewer for the suggestion. We modified the text in the revised manuscript.

Comment 5: Section 2.5 – The treatments are not clear (concentration, time of incubation..).

Response: Thanks to the reviewer for mentioning it. We mentioned the treatment concentrations and time in the revised manuscript.

Comment 6: Regarding the statistical test, no assessment of data normality was performed.

Response: We thank the reviewer for this comment. The normality of our datasets was assessed using the Shapiro–Wilk normality test, and all datasets passed this test (p > 0.05), mentioned in the method section line 240-242. Therefore, the use of Student’s t-test and one-way ANOVA with Duncan’s multiple comparison test was appropriate. This clarification has been added to the Methods section.

Comment 7: Figure 1C depicts the data from H/R-induced oxidative stress. The first bar refers to vehicle groups, but the viability is too low. No naïve group was used. Furthermore, all tested concentrations presented a similar effect, reinforcing the importance of a concentration-response curve.

Response: Thank you to the reviewer for the comment. Figure 1 represents neuronal viability in normal condition, not under H/R induced oxidative stress.

The first bar represents vehicle group in which viability is low as some cells die after the cells are seeded after primary neuronal culture. All tested concentrations show similar effect in case of viability as they don’t have toxic effect on neurons and also have mild antioxidant effects that preserve viability of the neurons.

As detailed in Supplementary Figure 1, we tested a wide range of concentrations and selected the three most effective for further experiments; under baseline conditions shown in Figure 1, these concentrations preserve viability without toxic effects.

Comment 8: Upon reviewing the results section, it became evident that the vast majority of the evaluations discussed lack proper documentation in the methodology (Sections 3.2 to 3.6). It is imperative to revise these sections to include a detailed account of the methods employed and to clarify the analytical approaches utilized.

Response: Thanks to the reviewer. As per reviewer’s suggestion we discussed the analysis approach in a detailed way in method section 2.8, line 172-192.

Comment 9: Chemical characterization should be the first data presented for logical order.

Response: The manuscript is re-arranged according to the reviewer’s suggestion.

Cooment 10: In the discussion, authors stated that an in vitro test was also performed with linalool. But it was not properly described in the method section.

Response: Thanks to the reviewer for mentioning it. We mentioned it in the method section line 136-137, and also the figure is represented in supplementary figure 1.

Comment 11: Figures are informative and of high quality.

Response:Thanks to the reviewer for the comment.

Comment 12: The discussion lacks depth and essentially repeats the findings without adequately linking them to relevant literature. A more comprehensive analysis is needed.

Additionally, it is recommended that the authors include a brief discussion on the potential limitations of their findings to allow for a more nuanced interpretation of the data.

Response: We edited the discussion with essential details  (line 500-546) and included the limitation of our study (line 535-546) accordingly in the reviewed manuscript.

Comment 1: The title fails to encapsulate the core findings of the manuscript accurately. The abstract is shallow and needs revision to include results consistently; upon reading this section, I find the potential applications of the findings unclear. Additionally, the study's objective is not clearly articulated in the abstract. Provide concrete findings instead of generally citing the results. No proper methods were described. Keywords should be revised to utilize terminology distinct from both the title and the abstract. Please, improve it.  

response:  Thank you for this insightful suggestion. We agree that the original title was too broad. To more accurately reflect our findings, we have revised the title to:
“Neuroprotective and Neurotrophic Potential of Flammulina velutipes Extracts in Primary Hippocampal Neuronal Culture.”

We have thoroughly revised the abstract to state the study objective clearly, describe the methods, present concrete results, and highlight the potential applications of our findings.

Flammulina velutipes (enoki mushroom) is a functional edible mushroom rich in antioxi-dants, polysaccharides, mycosterols, fiber, and minerals. Accumulating evidence high-lights its therapeutic potential across diverse pathological contexts, including boosting cognitive function. However, its role in neuromodulation has not been systematically ex-plored. This study examined the effects of methanolic and ethanolic extracts of F. velutipes on primary hippocampal neurons. Neurons were treated with different extract concentra-tions, followed by assessments of cell viability, cytoarchitecture, neuritogenesis, matura-tion, and neuroprotection under oxidative stress. The extracts were further characterized by GC-MS to identify bioactive metabolites, and molecular docking combined with MM-GBSA binding energy analysis was employed to predict potential modulators. Our results demonstrated that the methanolic extract significantly enhanced neurite out-growth, improved neuronal cytoarchitecture, and promoted survival under oxidative stress, whereas the ethanolic extract produced moderate effects. Mechanistic studies indi-cated that these neuroprotective and neurodevelopmental benefits are mediated through activation of the NTRK receptors, as validated by both in vitro assays and molecular docking studies. Collectively, these findings suggest that F. velutipes extracts, particularly the methanolic fraction, may serve as promising neuromodulatory agents for promoting neuronal development and protecting neurons from oxidative stress.”

Additionally, we have updated the keywords to better reflect the main findings while avoiding redundancy with the title and abstract.

Comment 2: Proper botanical identification must be provided to source the mushroom – “local market” is not an adequate source. The authors stated that “The mushrooms were characterized by expert personnel.”. Please, provide additional information about it.

Response:Thanks to the reviewer for raising the question. We added the modified version to section 2.2, line 85-92.

Comment 3: Still in section 2.2, there is an inadequate sentence: “Working extract of 8mg/ml was prepared. 30,60,100 μg/ml doses of both extracts were used in this study.” A concentration-response curve should be performed to comprehend the biological properties of the extract better. At least a rationale behind this concentration must be provided.

Response: According to the reviewer’s suggestion the concentration-response curve is represented in supplementary figure 1, showing why these concentrations were chosen. We initially applied concentrations ranged from 7. 5 to 100 ug/mL and chose the best three concentrations based on biological effect.

Comment 4: Revise all the text to replace the misuse of the word “dose” with “concentration,” considering the in vitro design of your study.

Response: Thanks to the reviewer for the suggestion. We modified the text in the revised manuscript.

Comment 5: Section 2.5 – The treatments are not clear (concentration, time of incubation..).

Response: Thanks to the reviewer for mentioning it. We mentioned the treatment concentrations and time in the revised manuscript.

Comment 6: Regarding the statistical test, no assessment of data normality was performed.

Response: We thank the reviewer for this comment. The normality of our datasets was assessed using the Shapiro–Wilk normality test, and all datasets passed this test (p > 0.05), mentioned in the method section line 240-242. Therefore, the use of Student’s t-test and one-way ANOVA with Duncan’s multiple comparison test was appropriate. This clarification has been added to the Methods section.

Comment 7: Figure 1C depicts the data from H/R-induced oxidative stress. The first bar refers to vehicle groups, but the viability is too low. No naïve group was used. Furthermore, all tested concentrations presented a similar effect, reinforcing the importance of a concentration-response curve.

Response: Thank you to the reviewer for the comment. Figure 1 represents neuronal viability in normal condition, not under H/R induced oxidative stress.

The first bar represents vehicle group in which viability is low as some cells die after the cells are seeded after primary neuronal culture. All tested concentrations show similar effect in case of viability as they don’t have toxic effect on neurons and also have mild antioxidant effects that preserve viability of the neurons.

As detailed in Supplementary Figure 1, we tested a wide range of concentrations and selected the three most effective for further experiments; under baseline conditions shown in Figure 1, these concentrations preserve viability without toxic effects.

Comment 8: Upon reviewing the results section, it became evident that the vast majority of the evaluations discussed lack proper documentation in the methodology (Sections 3.2 to 3.6). It is imperative to revise these sections to include a detailed account of the methods employed and to clarify the analytical approaches utilized.

Response: Thanks to the reviewer. As per reviewer’s suggestion we discussed the analysis approach in a detailed way in method section 2.8, line 172-192.

Comment 9: Chemical characterization should be the first data presented for logical order.

Response: The manuscript is re-arranged according to the reviewer’s suggestion.

Cooment 10: In the discussion, authors stated that an in vitro test was also performed with linalool. But it was not properly described in the method section.

Response: Thanks to the reviewer for mentioning it. We mentioned it in the method section line 136-137, and also the figure is represented in supplementary figure 1.

Comment 11: Figures are informative and of high quality.

Response:Thanks to the reviewer for the comment.

Comment 12: The discussion lacks depth and essentially repeats the findings without adequately linking them to relevant literature. A more comprehensive analysis is needed.

Additionally, it is recommended that the authors include a brief discussion on the potential limitations of their findings to allow for a more nuanced interpretation of the data.

Response: We edited the discussion with essential details  (line 500-546) and included the limitation of our study (line 535-546) accordingly in the reviewed manuscript.

Reviewer 2 Report

Comments and Suggestions for Authors
The study presented in the manuscript investigates the effects of two Flammulina velutipes alcoholic extracts (FVME – methanol and FVEE – ethanol) on the development and survival of hippocampal neurons. Both extracts stimulate neuronal viability, neurite outgrowth, and neuronal network formation. Still, FVME appears to exert a stronger effect on early and mature neuronal development, including extension and branching of axons and dendrites.
By GC-MS analysis, the authors identified 21or 22including linalool. Molecular docking authors write that the compounds (without specifying what type of molecules) bind efficiently to TrkA (NTRK1) and TrkB (NTRK2) receptors.

The article is very interesting, but in this form cannot be published.

The following revisions and improvements are needed as follows:

1) A sentence cannot begin with an abbreviation. Authors should read the manuscript carefully and review all sentences that begin with an abbreviation.
2) The pictures from the manuscript do not have real correspondence in the text.
2a) All figures that appear in the manuscript must also be found in the text, with the corresponding explanations (e.g., Figure 1a; 1b, 1c; Figure 21, 2b, 2c, etc.).
2b) The figures that appear in the description of the different paragraphs are incorrectly attributed. For example, Figure 5 is mentioned in paragraph 3.8, where in fact a reference should be made to Figure 7b.

3)Regarding molecular docking:
-Where is the figure 8b?
- what does C1; C2; C3....C40 represent?
-what type of ligand(s) were used for docking? (must be mentioned in the methodology used in molecular docking).
- Did the authors use a reference for docking? If so, which one?
- The GC-MS analysis is usually used for volatile organic compounds or gases. It is possible that in the set temperature range (200-280)oC the organic compounds in the crude extracts decompose, and the result actually contains the compounds resulting from decomposition (the authors can easily verify this by thermal analysis). If the authors did not have or do not have HPLC or LC/MS techniques andstandardized compounds, available for the two crude extract characterisation,  then in presenting the results the authors must refer (with bibliographic references) to the results obtained in other studies on the same type of macromycete, which to confirm the existence of the compounds identified by GC-MS techniques, such as linalool for example). Otherwise, the results presented in the molecular dockings are not supported.
-The binding energies from Figure 8a correspond to what type of ligand(s)? Here, the information can be represented as a table for a better understanding.
4) In what solvent were the crude extracts dissolved to obtain a solution of 8mg/mL? (must be mentioned in the methodology)
5) For GC-MS analysis, were the crude extracts dissolved in dimethylsulfoxide? (must be mentioned in the methodology)
6) The units of measurement must be written academically. For example, in the case of concentrations:  8mg/mL; 30µg/mL; 60µg/mL; 100µg/mL;  for temperature -20°C instead of -200c

Author Response

The study presented in the manuscript investigates the effects of two Flammulina velutipes alcoholic extracts (FVME – methanol and FVEE – ethanol) on the development and survival of hippocampal neurons. Both extracts stimulate neuronal viability, neurite outgrowth, and neuronal network formation. Still, FVME appears to exert a stronger effect on early and mature neuronal development, including extension and branching of axons and dendrites.
By GC-MS analysis, the authors identified 21or 22 including linalool. Molecular docking authors write that the compounds (without specifying what type of molecules) bind efficiently to TrkA (NTRK1) and TrkB (NTRK2) receptors.

The article is very interesting, but in this form cannot be published.

The following revisions and improvements are needed as follows:

Comment 1- A sentence cannot begin with an abbreviation. Authors should read the manuscript carefully and review all sentences that begin with an abbreviation.

Response 1-Thanks to the reviewer for pointing, it out. We corrected the manuscript as suggested.
Comment 2- The pictures from the manuscript do not have real correspondence in the text.
2a) All figures that appear in the manuscript must also be found in the text, with the corresponding explanations (e.g., Figure 1a; 1b, 1c; Figure 21, 2b, 2c, etc.).
2b) The figures that appear in the description of the different paragraphs are incorrectly attributed. For example, Figure 5 is mentioned in paragraph 3.8, where in fact a reference should be made to Figure 7b.

Response 2- Thanks to the reviewer for the comment. We correctly cited the figures in the text in the revised manuscript with corresponding explanation.

Comment 3- Regarding molecular docking:

-Where is the figure 8b?

Response-The figure 8b is now correctly included in the figure.

- what does C1; C2; C3....C40 represent?

Response-The compounds used for the molecular docking has now included in the Supplementary figure S2 with compound name for better understanding.

-what type of ligand(s) were used for docking? (must be mentioned in the methodology used in molecular docking).

- Did the authors use a reference for docking? If so, which one?

Response-Accoding to the reviewer’s suggestion the methodology of the docking section has beed edited with required details and references line 196-225.          
- The GC-MS analysis is usually used for volatile organic compounds or gases. It is possible that in the set temperature range (200-280)oC the organic compounds in the crude extracts decompose, and the result actually contains the compounds resulting from decomposition (the authors can easily verify this by thermal analysis). If the authors did not have or do not have HPLC or LC/MS techniques andstandardized compounds, available for the two crude extract characterisation,  then in presenting the results the authors must refer (with bibliographic references) to the results obtained in other studies on the same type of macromycete, which to confirm the existence of the compounds identified by GC-MS techniques, such as linalool for example). Otherwise, the results presented in the molecular dockings are not supported.

Response-Thanks to the reviewer for the question. The required references have been included in the manuscript line 517-519.

-The binding energies from Figure 8a correspond to what type of ligand(s)? Here, the information can be represented as a table for a better understanding.

Response-The figure 8a and 8b has been edited and the ligands have been presented clearly for better understanding. The binding energy and ligands detail has been included in Supplementary figure S2.

Comment 4-In what solvent were the crude extracts dissolved to obtain a solution of 8mg/mL? (must be mentioned in the methodology)

Response-Thanks to the reviewer for the comment. The information is added to the main text in revised manuscript, line 108-109.

Comment 5- For GC-MS analysis, were the crude extracts dissolved in dimethylsulfoxide? (must be mentioned in the methodology)

Response-Yes, for GC-MS analysis, the crude methanol and ethanol extracts of Flammulina velutipes were dissolved in dimethyl sulfoxide (DMSO). This information has now been included in the Materials and Methods section of the revised manuscript, line 116-117.

Comment 6- The units of measurement must be written academically. For example, in the case of concentrations:  8mg/mL; 30µg/mL; 60µg/mL; 100µg/mL;  for temperature -20°C instead of -200c

Response-The unit measurements are corrected in the revised manuscript.

Round 2

Reviewer 1 Report

Comments and Suggestions for Authors

My comments and suggestions were adequately addressed. The current version of the manuscript is suitable for publication.

Reviewer 2 Report

Comments and Suggestions for Authors

All my observation were implemented in the text, so now the manuscript is OK.